# Synthesis, Spectral Characterization and Biological Activities of Co(II) and Ni(II) Mixed Ligand Complexes

**DOI:** 10.3390/molecules26040823

**Published:** 2021-02-05

**Authors:** P. Manimaran, S. Balasubramaniyan, Mohammad Azam, D. Rajadurai, Saud I. Al-Resayes, G. Mathubala, A. Manikandan, S. Muthupandi, Zishan Tabassum, Imran Khan

**Affiliations:** 1PG & Research Department of Chemistry, Government Arts College, Ariyalur 621713, India; pmanimaran149@gmail.com (P.M.); balasubramaniyanchem@gmail.com (S.B.); 2Department of Chemistry, College of Science, King Saud University, P.O. Box 2455, Riyadh 11451, Saudi Arabia; sresayes@ksu.edu.sa; 3Department of Chemistry, Government Arts College (Autonomous), Karur 639005, India; rajaduraiavit@gmail.com; 4Department of Chemistry, Bharath Institute of Higher Education and Research, Selaiyur, Chennai 600073, India; madhu2705@gmail.com or; 5Department of Physics, Loyola College, Affiliated to University of Madras, Chennai 600034, India; muthupandisankar@gmail.com; 6Department of Chemistry, Rajshree Institute of Management and Technology, Bareilly 243122, India; zishant@gmail.com; 7Applied Science Humanities Section, University Polytechnic, Faculty of Engineering and Technology, Aligarh Muslim University, Aligarh 202002, India; imrannano@gmail.com

**Keywords:** mixed-ligand complexes, DNPH, antimicrobial, DPPH assay

## Abstract

2,4-Dinitrophynylhydrazine and two thiocyanate ions in a (M:L1:L2) 1:2:2 molar ratio was synthesized in the complexes of Co(II) and Ni(II). The prepared compounds were identified through a C.H.N.S. analysis, conductivity measurements, powder X-ray diffraction (PXRD), the infrared spectrum, and a UV-visible spectrum analysis, in addition to the magnetic properties being measured. The measurements of the molar conductance implieda nonelectrolytic nature of compounds Co(II) and Ni(II). The magnetic susceptibility, as well as electronic spectra, represented all the metal complexesthroughoctahedral geometry, respectively. The PXRD patterns suggested that all the complexes were an orthorhombic system with unit cell parameters. The in-vitro biological activity of the ligand and the metal complexes were screened against the Gram-positive and negative pathogenic bacteria Staphylococcus aureus, Bacillus subtilis, *Pseudomonas*, *aeruginosa* and *Escherichia coli*, as well as the fungal species of *Aspergillusniger* and *Candida albicans*.Thus, the metal complexes showeda high efficiency of antimicrobial activity compared with the ligand. Furthermore, applications of the ligand, as well as the metal complexes, were tested for in-vitro antioxidant potential in aDPPH assay. The results showed that the activity of the metal complexes with the in-vitro antioxidant was more active than that of 2,4-dinitrophenylhydrazine(DNPH).

## 1. Introduction

Mixed chelating metal complexes are the focus for lots of researchers, as theyaremore than ever hugely important within the biological organism in pharmacological applications like antibacterial and fungal, anticancer, anti-inflammatory and antitumor [1]. The coordination chemistry of transition metal ions through different forms of ligands has been improved through the recent developments within the fieldof bioinorganics, as well as medicinal chemistry [2]; transition metals have a significant role to play in utilizing transition metal complexes as drugs for treatments for many diseases, which is in an important field of research [3]. As a probe of the biological system, steady and harmless metal complexes among active metal places are valuable [4]. The environment of the metal center, such as coordination geometry, ligands and, also, donor groups, are the key factors for metalloproteins performing unique physiological functions [5]. 2,4-Dinitrophenylhydrazine (DNPH) is a vital part of various biological and pharmaceutical applications [6]. It is an analog of H_2_NNH_2_ (hydrazine) and, moreover, an essential class of drugs [7]. Nitrogen and, furthermore, oxygen are able to be coordinated, which is helpful for bioinorganic processes [8]. The SCN (thiocyanate) faction can be used as a negative ion; otherwise, one can use mono-dentate chelating coordinates like S and N with a bridging ligand. The high coordination capability of the SCN group has a variety of bonds, respectively [9].

This current report synthesized the compounds of Co(II)and Ni(II) from DNPH and SCN ions and, furthermore, the physicochemical characterizations, and the goal was to study the biological potential of DNPH among the complexes of Co(II) and Ni(II).

## 2. Materials and Methods

### 2.1. Materials

All the chemicals were purchased from commercial sources and were used with no further purification process (CoCl_2_·6H_2_O, NiCl_2_·6H_2_O and KSCN) from Merck, and 2,4-dinitrophenylhydrazone was obtained from Sigma Aldrich, along with the solvents like DMSO, DMF, acetone, ethanol and methanol, used as AnalaR grade.

### 2.2. Analytical Characterizations

An elemental analysis was performed by using a CHNS analyzer. Conductivity measurements were reported from (Equip–Tronics, eq-661A) a conductivity meter at 30 °C in DMF medium (10^−3^ M). The infrared spectrum measured the wavelength from 4000–400 cm^−1^.The electronic absorption spectra were carried out on Jasco V-630 optical resolution 1 nm and wavelength ranging from 200–800 nm, and the powder X-ray diffraction (PXRD) patterns were carried out on a Shimadzu. The magnetic properties were measured from the Model MsB-MK1 using Guey’s Balance Magnetic Susceptibility.

### 2.3. Synthesis of Co(II) and Ni(II) Mixed Ligand Complexes

The complexes of Co(II) and Ni(II) were prepared by DMF solution of (20 mM) DNPH with an ethanol solution of metal chloride (10 mM). The above solution was magnetically stirred and refluxed about one hour, as well as adding an addition of 20-mM de-ionized water solution of potassium thiocyanate (Figure 1). The entire solution was refluxed at six hours. Finally, the colored complex was cooled, filtered in addition to washed with EtOH solution and, after that, dried out in anhydrous CaCl_2_ vacuum desiccators.

### 2.4. In-Vitro Antimicrobial Studies

DNPH and the complexes of Co(II) and Ni(II) were treated for antimicrobial efficiency via using various species of Gram-positive and, furthermore, Gram-negative pathogenic bacteria and fungi, such as *B. subtilis*, *E. coli*, *P. aeruginosa*, *S. aureus*, *A. niger* and *C. albicans*, using the disc diffusion method. The free ligand, as well as Co(II) and Ni(II) compounds, were dissolved in DMSO. The variations (30, 60 and 90 µg/mL) of the solution were arranged individually. The prepared discs were dipped within a specified variation of free ligand, in addition to the complexes located within petri plates containing a nutrient medium sowed through every bacterial and fungal serum individually. Every plate was incubated for 24 and 48 h for the bacteria, as well as fungi, at 37 °C; furthermore, the zone of inhibition values were noted [10].

### 2.5. In-Vitro Antioxidant (DPPH) Assay

DPPH has generally been used to estimate the free radical scavenging ability of different antioxidants. The radical scavenging activity of metal complexes and ligand-tested samples were determined with the (2,2-diphenyl-1-picrylhydrazyl) DPPH technique [11]. The entire tested samples were prepared in various combinations (50, 100 and 150 µg/mL) of metal complexes, and free ligand-blended with 2-mL MeOH (0.1 mM) solution of the DPPH radical and kept at room temperature for 30 min in a dark place. Then, the decreased absorption was measuring at 517 nm in the UV-visible spectrum. The % of radical scavenging efficiency was considered from the following equation [12]:DPPH Scavenging ability(%) = Abs control − Abs sample/Abs control × 100(1)

## 3. Results and Discussion

The physical properties with the analytical data of the metal complexes are summarized in Table 1. The Co(II) and Ni(II) complexes are stable at room temperature and soluble in DMSO and DMF. The CHNS analysis of the ligand, as well as metal compound, experimental values are in good agreement with the theoretical calculated values with the complex ratio 1:2:2 between the metals and ligand.

### 3.1. Molar Conductance

The molar conductance values of the metal complexes were measured at a 1 × 10^−3^ M concentration of DMF solution, and all the complexes showed conductance in the range of 17.86–19.85 Ω^−1^ cm^2^ mol^−1^ at 37 °C, as shown in Table 1. The lowest conductivity value indicated a nonelectrolyte nature, and outside their coordination sphere, there is no counter ion present in the complexes [13].

### 3.2. FTIR Spectra Analysis 

The infrared spectra analyses of ligands 2,4-dinitrophenylhydrazine and thiocyanate and their corresponding coordination compounds (Figure 2) were carried out, and the relevant peaks (cm^−1^) are given in Table 2.

The FTIR spectrum of the ligands (DNPH) and their metal complexes were characterized at bands 3324, 1319 and 920 cm^−1^, assigned to ν(NH) amine and aromatic nitro ν(NO), in addition to ν(N–N) hydrazine, respectively, for DNPH [14]. In the IR spectrum, the complexes showed evidence of ligand bands through suitable shifts due to their complex formations [15]. The IR spectra of 2,4-dinitrophenyl hydrazine observed sharp peaks at 3324 cm^−1^ due to the ν(N-H) frequency, which was moved to a lower frequency at 3288–3278 cm^−1^ for the metal complexes, thus signifying that the amino nitrogen group is coordinated to a metal atom. Further supporting the observed ν(N-N) stretching frequency, it was shifted to a higher wave number in the metal complexes [16]. The ν(NO) symmetric stretching band was indicated at 1319 cm^−1^; furthermore, these band were shift higher at 53–38 cm^−1^, and there was a downward change of the ν(NO_2_) asym to 15–20 cm^−1^ in the spectra of the complexes. These findings indicate that the ligands were coordinated with metal ions through the one oxygen of the nitro group [17]. The new bands in the complexes were about 2072–2095 cm^−1^ due to the ν(SCN). These results showed that thiocyanate coordinated with the metal ions through the nitrogen atom bonded to NCS in iso-thiocyanate mode [18,19,20]. The lower frequency bands appeared in two synthesized complexes at 539–551 cm^−1^, in addition to 412–496 cm^−1^, which was attributed to the ν(M-O), as well as ν(M-N) bands, correspondingly [21]. The FTIR spectra results showed that the ligands were coordinated with metal ion via amino N, nitro O and SCN nitrogen atoms engaging coordination sides in the complexes.

### 3.3. UV-Visible Spectrum and Magnetic Moments

The electronic absorption spectrum of DNPH and their Co(II) and Ni(II) complexes in DMF are shown in Figure 3, which contains the absorption regions, band assignments and the proposed geometry of the complexes given in Table 3.

2,4-Dinitrophenyl hydrazine exhibits two absorptions at 363 nm and 267 nm, assignable to n→π* and π→π*, correspondingly.

The Co(II) complex absorption bands at 281, 388 and721 nm are attributed to the π→π*, LMCT and ^4^T_1g_(F)→^4^T_2g_(F) transitions, signifying an octahedral geometry. A measured magnetic property assessment of 5.1 BM may be additional evidence for octahedral geometry [22].

The Ni(II) complex provides absorption bands at 284, 390 and 444, as well as 724 nm, are attributed to the π→π*,LMCT, ^3^A_2g_(F)→^3^T_1_g(P) and ^3^A_2g_(F)→^3^T_1g_(F)transitions, respectively, corresponding to a high spin octahedral geometry. Further confirmation was achieved by the magnetic moment range at 3.2 BM, which is consistent for the suggested octahedral arrangement [23,24].

### 3.4. Powder XRD Analysis

The powder X-ray diffraction analysis of the Co(II) and Ni(II) complexes are given in Figure 4. The PXRD pattern of the metal complexes was defined as sharp crystalline peaks. The PXRD pattern of the metal complex lattice parameters were calculated through the assistance of a computer program, XPERT PRO. The metal complexes gave the values of the lattice constants a = 10.82, α = 900, b = 9.24(4), β = 900, c = 6.03(2), γ = 900, V = 65.25 and the crystal system orthorhombic and Bravais-type Primitive (P) for the Co(II) complex. The Ni(II) complex lattice constants are a = 11.14, α = 90°, b = 9.59(4), β = 90°, c = 4.79(2), γ = 90°, V = 511.05 and the crystal system of orthorhombic and Bravais type Primitive (P). The crystallite sizes were calculated for the compounds of Co(II) and Ni(II) using Debye Scherrer’s Equation [25]:D = 0.9λ/β cosθ(2)
where λ = wavelength of the X-ray radiation (Cu Kα = 1.5406 A°), β = (FWHM) full-width half-maximum, θ = diffraction angle and constant as 0.9. The crystallite sizes were around 29.28 and 30.59 nm, respectively.

### 3.5. Antimicrobial Efficiency

The antimicrobial activity of the ligands, in addition to the Co(II) and Ni(II) complexes, against the pathogenic bacteria and fungi activity results are shown in Table 4, along with Figure 5 The antimicrobial results show that the ligand is fairly active; furthermore, the metal compounds have a better efficiency than the ligand. The Co(II) complex is extremely active against the bacteria and fungi species compared with the Ni(II) complex and free ligand.

The toxicity functions of the compounds and ligand can be due to an improvement in the lipophilic nature. The polarity of metal atoms is primarily decreased by the ligand due to the limited distribution of the positive charge through the donor groups and potential delocalization of the π–electron. The ligand also enhances the central metals’ lipophilic natures, which consequently favors permeation through the cell membrane lipid layer [26,27,28]. For a raise in the concentrations of the compounds, the activity increases. It is proposed that complexes with antimicrobial potential will also perform in destroying the microorganism or otherwise preventing microbe reproduction by blocking its active sites [29]. The complexes’ efficiency is arranged as follows: Co(II) > Ni(II) > L.

### 3.6. Antioxidant Studies (DPPH Assay)

The increased antioxidant activity of the complexes is able to be ascribed to an electron withdrawing effect of the metal ions that facilitate letting loose theH^+^ atom to decrease the DPPH radical. Under certain atmospheric conditions, and during normal cellular function in the body, free radicals are generated. Thus, antioxidants play an important role in preserving the human body from the harm of reactive oxygen species [30]. The antioxidant activities of 2,4-dinitrophenylhydrazine and its Ni(II) and Co(II) compounds were studied with DPPH, as seen in Figure 6, as well as Table 5. The Ni(II) complex exhibited strong scavenging efficiency, whereas the Co(II) complex exhibited more moderate activity than the ligand. Compare to the ligand, the higher scavenging behaviors of the Ni(II) complexes can be attributed to the coordination of metal through the ligand [31].

## 4. Conclusions

2,4-Dinitrophenylhydrazine and the thiocyanate complexes of Co(II) and Ni(II) were synthesized and characterized. The C, H, N and S analyses showed that the complexes have a 2:1ratio of ligand to metal. The evidence on the molar conductivity revealed that the compounds are non electrolytic in nature. The octahedral structure for the complexes of Co(II) and Ni(II) was determined based on the results obtained from the UV-visible spectral study and magnetic susceptibility measurements. The powder XRD analysis revealed that the compounds are average crystalline sizes are around 29.28 and 30.59 nm. The in-vitro antibacterial and fungal efficiency pointed out that the complexes of Co(II) and Ni(II) have superior action to DNPH. The in-vitro antioxidant efficiency revealed that both complexes had higher activity than DNPH.

## Figures and Tables

**Figure 1 molecules-26-00823-f001:**
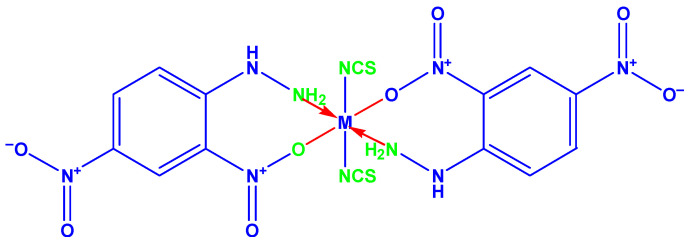
Structures of the metal complexes (M = Co(II) and Ni(II)).

**Figure 2 molecules-26-00823-f002:**
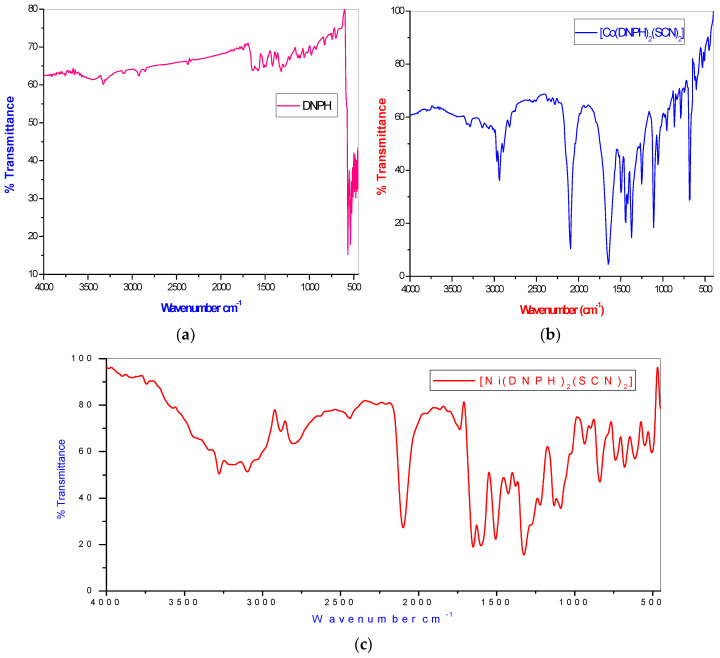
FTIR (Fourier transform infrared) spectra of (**a**) 2,4-dinitrophenylhydrazine (DNPH), (**b**) the Co(II) complex and (**c**) the Ni(II) complex.

**Figure 3 molecules-26-00823-f003:**
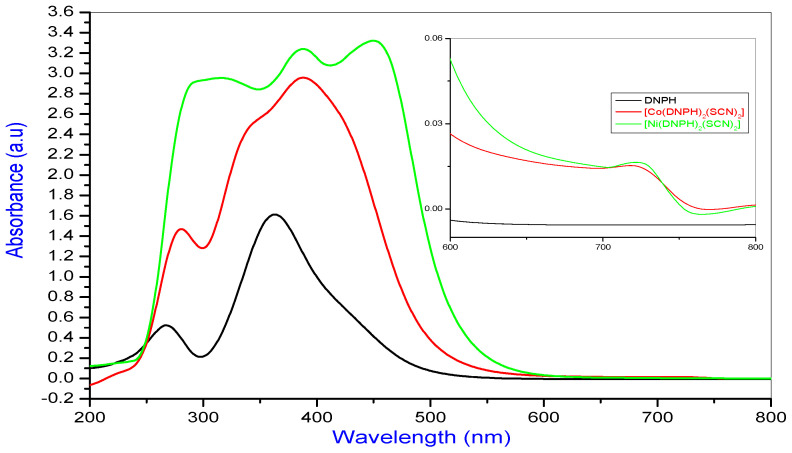
UV-visible spectrum of DNPH, along with the Co(II) and Ni(II) complexes.

**Figure 4 molecules-26-00823-f004:**
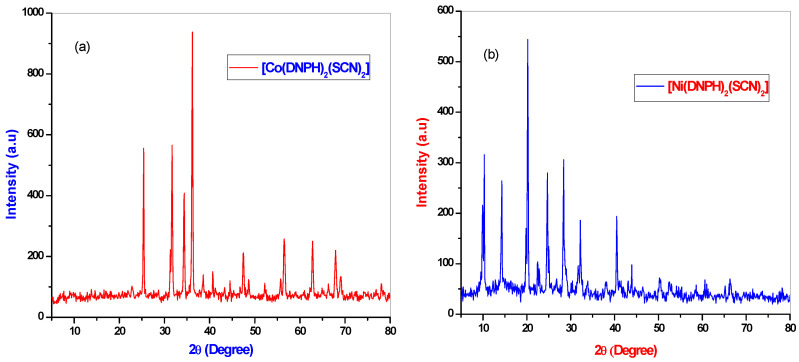
Powder X-ray diffraction (PXRD) patterns of the (**a**) Co(II) complex and (**b**) Ni(II) complex.

**Figure 5 molecules-26-00823-f005:**
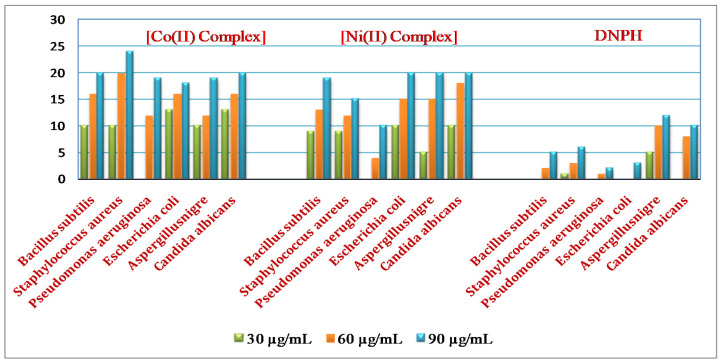
Results of the antimicrobial potential screening for DNPH and its metal complexes.

**Figure 6 molecules-26-00823-f006:**
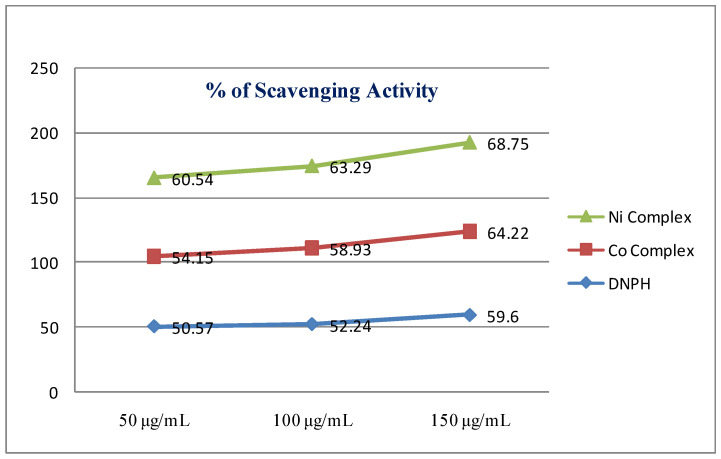
Result of the DPPH radical scavenging abilities.

**Table 1 molecules-26-00823-t001:** Physical data, as well as analytical characterizations, of the metal compounds.

Complexes	M.F	MW	Color of the Complexes	% of Yield	Elemental Analysis Found/(Calculated) %					Λ_m_ Ω^−1^cm^2^mol^−1^
					%C	%H	%N	%S	% Metal	
(Co(DNPH)_2_(SCN)_2_)	CoC_14_H_12_N_10_O_8_S_2_	571.36	Blue	69	29.45	2.12	24.52	11.22	10.31	19.85
					29.42	2.09	24.48	11.18	10.29	
(Ni(DNPH)_2_(SCN)_2_)	NiC_14_H_12_N_10_O_8_S_2_	571.12	Green	74	29.44	2.12	24.53	11.23	10.28	17.86
					29.37	2.08	24.47	11.22	10.20	

Λm—Molar conductivity; M.F—Molecular Formula; MW—Molecular Weight.

**Table 2 molecules-26-00823-t002:** Selected FTIR spectral data of the ligands and their metal complexes (cm^−1^).

Compounds	ν(N-H)	ν(NO_2_)asym	ν(NO_2_)asym	ν(N-N)	ν(SCN)	ν(M-O)	ν(M-N)
DNPH	3324	1516	1319	920	-	-	-
(Co(DNPH)_2_(SCN)_2_)	3284	1495	1371	969	2094	539	496, 446
(Ni(DNPH)_2_(SCN)_2_)	3278	1501	1378	978	2098	551	449, 423

**Table 3 molecules-26-00823-t003:** UV-visible absorption spectral data and magnetic moment values.

Complex	λ_max_ (nm)	Band Assignments	Geometry	Magnetic Moment (B.M)
DNPH	267	π–π*	-	-
	363	n–π*		
(Co(DNPH)_2_(SCN)_2_)	281	π–π*	Octahedral	5.1
	388	LMCT		
	721	4T1g(F)–4T2g(F)		
(Ni(DNPH)_2_(SCN)_2_)	284	π–π*	Octahedral	3.2
	390	LMCT		
	450	3A2g (F)→3T2g (P)		
	724	3A2g (F)→3T2g (F)		

LMCT—ligand to metal charge transfer.

**Table 4 molecules-26-00823-t004:** Antimicrobial activities of DNPH and their metal complexes.

Organism Name	Co(II) Complex			Ni(II) Complex			2,4-Dinitrophenyl Hydrazine		
Test (mm)	30 µg/mL	60 µg/mL	90 µg/mL	30 µg/mL	60 µg/mL	90 µg/mL	30 µg/mL	60 µg/mL	90 µg/mL
*Bacillus subtilis*	10	16	20	9	13	19	-	2	5
*Staphylococcus aureus*	10	20	24	9	12	15	1	3	6
*Pseudomonas aeruginosa*	-	12	19	-	4	10	-	1	2
*Escherichia coli*	13	16	18	10	15	20	-	-	3
*Aspergillusnigre*	10	12	19	5	15	20	5	10	12
*Candida albicans*	13	16	20	10	18	20	-	8	10

**Table 5 molecules-26-00823-t005:** Percent of radical scavenging abilities of the Co(II) and Ni(II) complexes and DNPH.

Compounds	150 μg/mL	100 μg/mL	50 μg/mL
2,4-dinitrophenyl hydrazine	59.60	52.24	50.57
Co(II) complex	64.22	58.93	54.15
Ni(II) complex	68.75	63.29	60.54

## Data Availability

Not applicable.

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
