# Peer review of "Synthesis, Spectral Characterization and Biological Activities of Co(II) and Ni(II) Mixed Ligand Complexes"

_molecules, 2021, doi:10.3390/molecules26040823_

Round 1

Reviewer 1 Report

The manuscript ''SYNTHESIS, SPECTRAL CHARACTERIZATION AND BIOLOGICAL ACTIVITIES OF CO(II), AND NI(II) MIXED LIGAND COMPLEXES'' can be considered to be accepted for the publication after following changes:

  • English must be completely improved;
  • Introduction should be rewritten, with more details of metal complexes in the fields of bioinorganic and medicinal chemistry. And then special attention should be focused on the topic described in this paper (type of ligand, metal centers, etc.);
  • Scheme which represents synthesis of the complexes, including structures of ligand and obtained complexes should be presented;
  • All characterization data should be added in manuscript;
  • Complexes need to be characterized by NMR and MS analysis, as well.

Author Response

Reviewer 1: Comments and Suggestions for Authors

The manuscript ''SYNTHESIS, SPECTRAL CHARACTERIZATION AND BIOLOGICAL ACTIVITIES OF CO(II), AND NI(II) MIXED LIGAND COMPLEXES'' can be considered to be accepted for the publication after following changes:

  • English must be completely improved;

Ans; Improved

  • Introduction should be rewritten, with more details of metal complexes in the fields of bioinorganic and medicinal chemistry. And then special attention should be focused on the topic described in this paper (type of ligand, metal centers, etc.);

Ans: Details given introduction page No.2

  • Scheme which represents synthesis of the complexes, including structures of ligand and obtained complexes should be presented;

Ans: Scheme given in Page  No. 2

  • All characterization data should be added in manuscript;

Ans: Added

  • Complexes need to be characterized by NMR and MS analysis, as well.

Ans: The authors are apologised, due to this pandemic situation we are not able to characterized NMR and MS analysis. In future we will do that.

Reviewer 2 Report

Manimaran and colleagues report two novel transition metal complexes incorporating 2,4-DNP. The work is generally well done but the English is poor throughout as exemplified by sentences from the abstract:

“2,4-dinitrophynylhydrazine and thiocyanate ion in (M: L1: L2) 1:2:2 molar ratio was synthesized in the complexes of Co(II) and Ni(II).” should read: “Metal complexes formed from  Co(II) or Ni(II), 2,4-dinitrophenylhydrazine and thiocyanate in a 1:2:2 molar ratio were synthesized.”

“Thus, the metal complexes have shown high efficiency of antimicrobial activity compares with a ligand.” should read: “The metal complexes have shown high antimicrobial activity compared to the free DNPH ligand.”

In lines 45 and 46, what do the authors mean by: “Nitrogen furthermore oxygen be able to be concerned into coordination given that a helpful for bio-inorganic processes”?

2,4-DNPH is not, as stated in line 45, an analogue of hydrazine; the only thing they have in common is a terminal N-NH2 group.

The syntheses and analyses, together with the antimicrobial experiments, have been undertaken using appropriate methods.

The conclusions concerning the geometry, spin state, etc of the products are reasonable, given the data presented, but a figure illustrating their proposed structures would be useful for readers wishing to visualize them. In particular, how do they envisage the 2,4-DNP to act as a bidentate ligand. Based on the symmetry from the IR spectra, do they expect the octahedral complexes to be cis or trans isomers? The authors should also make some comparison with structural and antimicrobial data for similar 2,4-DNPH in the literature, e.g. Platinum and palladium complexes with 2,4-dinitrophenylhydrazine: Synthesis and cytotoxic activity, de Souza et al., Latin Am. J. Pharm., 2012, 31, 620-624; and 2,4-DNP derivative complexes tested against five microbial species in: Complexes of cobalt(II), nickel(II) and zinc(II) with Schiff bases derived from 4-anisaldehyde, Ndahi and Nasiru, IJPSR, 2012, 3, 5116-5120.

Once these issues have been addressed – including correcting the English throughout - the work will be suitable for publication in Molecules.

Author Response

Reviewer 2: Comments and Suggestions for Authors

Manimaran and colleagues report two novel transition metal complexes incorporating 2,4-DNP. The work is generally well done but the English is poor throughout as exemplified by sentences from the abstract:

Ans: Improved.

 “2,4-dinitrophynylhydrazine and thiocyanate ion in (M: L1: L2) 1:2:2 molar ratio was synthesized in the complexes of Co(II) and Ni(II).” should read: “Metal complexes formed from  Co(II) or Ni(II), 2,4-dinitrophenylhydrazine and thiocyanate in a 1:2:2 molar ratio were synthesized.”

 ANS: Corrected. Abstract page No.1

“Thus, the metal complexes have shown high efficiency of antimicrobial activity compares with a ligand.” should read: “The metal complexes have shown high antimicrobial activity compared to the free DNPH ligand.”

 ANS: Corrected. Abstract page No.1

In lines 45 and 46, what do the authors mean by: “Nitrogen furthermore oxygen be able to be concerned into coordination given that a helpful for bio-inorganic processes”?

 ANS: Corrected line 45 and 46, introduction page No.2

2,4-DNPH is not, as stated in line 45, an analogue of hydrazine; the only thing they have in common is a terminal N-NH2 group.

 ANS: Corrected, introduction page No.2

The syntheses and analyses, together with the antimicrobial experiments, have been undertaken using appropriate methods.

The conclusions concerning the geometry, spin state, etc of the products are reasonable, given the data presented, but a figure illustrating their proposed structures would be useful for readers wishing to visualize them. In particular, how do they envisage the 2,4-DNP to act as a bidentate ligand. Based on the symmetry from the IR spectra, do they expect the octahedral complexes to be cis or trans isomers? The authors should also make some comparison with structural and antimicrobial data for similar 2,4-DNPH in the literature, e.g. Platinum and palladium complexes with 2,4-dinitrophenylhydrazine: Synthesis and cytotoxic activity, de Souza et al., Latin Am. J. Pharm., 2012, 31, 620-624; and 2,4-DNP derivative complexes tested against five microbial species in: Complexes of cobalt(II), nickel(II) and zinc(II) with Schiff bases derived from 4-anisaldehyde, Ndahi and Nasiru, IJPSR, 2012, 3, 5116-5120.

Ans: Draw the structure of complex, conclusion page No.2

Once these issues have been addressed – including correcting the English throughout - the work will be suitable for publication in Molecules.

Ans; Corrected

Reviewer 3 Report

This manuscript contains synthesis of 2,4-dinitrophenylhydrazine and thiocyanate complexes of Co(II) and Ni(II) and characterization of their structural and physical properties using C, H, N, S analyses, conductivity measurement, PXRD, infrared spectrum, UV-visible spectrum analysis in addition to magnetic properties. The background as well as the results are well presented and the authors take great care to limit biological potential of DNPH–Co(II) and Ni(II) complexes.Thus, this manuscript could be interesting for the readers in the field of inorganic chemistry as well as biological chemistry. The results could well serve as the basis for devel­opment of these fields. However, there are still lack of originality and novelty as well as experimental results.

Minor comments

  1. The authors should pay some comments to explain clearly the reason of choosing Co(II) and Ni(II) complexes with 2,4-dinitrophenylhydrazine and thiocyanate in other various metal ions.
  2. It is helpful for readers to show the octahedral structure for complexes of Co(II) and Ni(II) in the text. The authors should confirm the 1H NMR and 13C NMR spectroscopy and the single crystal measurements to support the octahedral structures for complexes of Co(II) and Ni(II). These are serious problems in this manuscript for publication.
  3. The authors should describe the melting point of the synthesized complexes in the experimental section.
  4. The authors should improve the typos, hyphenation and capital letters (many) in the text.

Thus, this manuscript should not be published in Molecules at the present stage. But after major revisions this manuscript might be acceptable in this journal.

Author Response

Reviewer 3: Comments and Suggestions for Authors

This manuscript contains synthesis of 2,4-dinitrophenylhydrazine and thiocyanate complexes of Co(II) and Ni(II) and characterization of their structural and physical properties using C, H, N, S analyses, conductivity measurement, PXRD, infrared spectrum, UV-visible spectrum analysis in addition to magnetic properties. The background as well as the results are well presented and the authors take great care to limit biological potential of DNPH–Co(II) and Ni(II) complexes.Thus, this manuscript could be interesting for the readers in the field of inorganic chemistry as well as biological chemistry. The results could well serve as the basis for devel­opment of these fields. However, there are still lack of originality and novelty as well as experimental results.

Minor comments

  1. The authors should pay some comments to explain clearly the reason of choosing Co(II) and Ni(II) complexes with 2,4-dinitrophenylhydrazine and thiocyanate in other various metal ions.

Ans:

  • Co(II) and Ni(II) metal ions are present in several inorganic pharmaceuticals used as drugs against a variety of diseases, ranging from antibacterial and antifungal to anticancer applications
  • SCN’s diverse properties as both host defences and antioxidant agent make it a potentially useful therapeutic.
  • A number of hydrazine derivatives have been widely used in the medical and industrial fields
  1. It is helpful for readers to show the octahedral structure for complexes of Co(II) and Ni(II) in the text. The authors should confirm the 1H NMR and 13C NMR spectroscopy and the single crystal measurements to support the octahedral structures for complexes of Co(II) and Ni(II). These are serious problems in this manuscript for publication.

Ans: added the octahedral structure of complexes (page No.8). The structure was determined on result obtained from Elemental analysis, FT-IR, UV-Visible, magnetic susceptibility and molar conductivity measurements.   

  1. The authors should describe the melting point of the synthesized complexes in the experimental section.

Ans: Corrected

  1. The authors should improve the typos, hyphenation and capital letters (many) in the text.

Ans: Corrected

Thus, this manuscript should not be published in Molecules at the present stage. But after major revisions this manuscript might be acceptable in this journal.

Round 2

Reviewer 1 Report

The quality of manuscript has been been improved and I would recommend to be accepted for publication.

Reviewer 3 Report

The authors have improved the original manuscript by following the referee’s comments. This manuscript could be interesting for the readers in the field of inorganic chemistry as well as biological chemistry.

I recommend this manuscript for publication in Molecules at the present stage.